# Artisanal fish fences pose broad and unexpected threats to the tropical coastal seascape

Dan A. Exton [1], Gabby N. Ahmadia[2], Leanne C. Cullen-Unsworth [3,4], Jamaluddin Jompa[5], Duncan May[6], Joel Rice[7], Paul W. Simonin[8], Richard K.F. Unsworth [4,9] & David J. Smith[10]

Gear restrictions are an important management tool in small-scale tropical fisheries, improving sustainability and building resilience to climate change. Yet to identify the management challenges and complete footprint of individual gears, a broader systems approach is required that integrates ecological, economic and social sciences. Here we apply this approach to artisanal fish fences, intensively used across three oceans, to identify a previously underrecognized gear requiring urgent management attention. A longitudinal case study shows increased effort matched with large declines in catch success and corresponding reef fish abundance. We find fish fences to disrupt vital ecological connectivity, exploit > 500 species with high juvenile removal, and directly damage seagrass ecosystems with cascading impacts on connected coral reefs and mangroves. As semi-permanent structures in otherwise open-access fisheries, they create social conflict by assuming unofficial and unregulated property rights, while their unique high-investment-low-effort nature removes traditional economic and social barriers to overfishing.

[1] Operation Wallacea, Wallace House, Old Bolingbroke, Spilsby, Lincolnshire PE23 4EX, UK. [2] Oceans Conservation, World Wildlife Fund, 1250 24th Street NW, Washington, DC 20037, USA. [3] Sustainable Places Research Institute, Cardiff University, 33 Park Place, Cardiff CF10 3BA, UK. [4] Project Seagrass, 33 Park Place, Cardiff CF10 3BA, UK. [5] Graduate School, Hasanuddin University, Makassar, South Sulawesi, Indonesia. [6] Rientraid, Drumbeg Road, Near Kylesku, By Lairg, Sutherland IV27 4HP, UK. [7] Rice Marine Analytics, 1690 Hillcrest Avenue, Saint Paul, MN 55105, USA. [8] Department of Ecology and Evolutionary Biology, Cornell University, Ithaca, NY 14853, USA. [9] Seagrass Ecosystem Research Group, College of Science, Swansea University, Swansea SA2 8PP, UK. [10] Coral Reef Research Unit, School of Biological Sciences, University of Essex, Colchester, Essex CO3 4SQ, UK. Correspondence and requests for materials should be addressed to D.A.E. (email: dan.exton@opwall.ac.uk)

The long-term future of tropical coastal ecosystems and their fisheries will largely be determined by our ability and willingness to address global climate change[1–6]. Yet, addressing localised threats (e.g., unsustainable or damaging fishing practices) can build short-term ecosystem resilience[7], improve adaptive capacity of species and ecosystems to changes in climate[8], and increase the opportunity for adaptation and acclimation[9]. However, there are potentially catastrophic impacts in areas where dependence on marine ecosystem services is high. For example, coral reefs provide the majority of protein to over 400 million people[10], yet an estimated two-thirds of reef fish are already believed to have been lost[11]. Developing solutions to slow the degradation of these ecosystems, including identifying the most-damaging practices and understanding what drives them, is therefore among the biggest global challenges facing humanity[12].

A range of tools are currently used for sustainable coastal fisheries management. Marine Protected Areas often provide a governance structure for tailored conservation action such as fisheries management and can successfully buffer ecological communities against disturbances[13–15], especially when optimal reserve size thresholds are met[16], manpower and funding are sufficient[17], and ecological connectivity is considered[18,19]. Where strict no-take rules are impractical, fish biomass can be maintained via more tailored fisheries management, including restrictions on gear usage and access[20,21], while strengthening fisheries governance at the local-scale best protects ecological processes[22]. Governance of fishing activities is made more challenging by the multi-gear nature of many tropical small-scale fisheries; it is not uncommon for over 50 distinct gear types to be in use at a single location (including multiple varieties of the same gear class)[23–26]. Although on occasion, critical management targets are easily identified, for example, blast fishing with explosives that has received considerable conservation attention[27,28], it is often less obvious where effort would be best spent to maximise conservation benefits. Similar to management interventions[29], individual gear types express characteristics that span ecological, economic, and social sciences, and to fully understand their true impact they should be viewed beyond the narrow perspective of whether they are visibly destructive. Instead, a multi-directional approach is needed to realise the complete footprint on both the human communities and ecological systems in which they are used.

Artisanal fishing gears are often promoted by managers, based on a widely held perception of smaller footprints compared with more-industrialised or modernised techniques. This is often despite the lack of an holistic understanding of the wide-ranging and far-reaching impacts these gears might pose. Artisanal fish fences (artisanal weirs, static fyke nets), for example, are used frequently and intensively, and their impacts appear severe and far-reaching, making them an ideal gear to test our approach. We show fish fences being used intensively across three ocean basins (Atlantic, Indian, Pacific), before presenting a unique 15-year case study incorporating ecological, fisheries catch and socioeconomic data to examine fish fence impacts across communities (natural and human) and ecosystems (seagrass, coral reef and mangrove), as well as exploring the social and economic drivers underpinning their use. Through such a comprehensive assessment, we are able to highlight otherwise unrecognised threats from this widely used gear type to local stakeholders and across multiple ecosystems, and demonstrate the value of a multi-disciplinary approach in directing evidence-based gear restriction efforts in tropical small-scale fisheries.

## Results and discussion

**Spatial and temporal patterns in usage and effort.** Artisanal fish fences are semi-permanent structures positioned on intertidal and shallow subtidal flats (Fig. 1a) that use fences to funnel fish into a holding structure at the seaward end as the water recedes towards low tide[23]. They were a common gear type globally before the advent of industrial fishing[30], and they remain in use across a wide geographical range as a component of small-scale tropical coastal fisheries. We report the use of artisanal fish fences from the published scientific literature in 19 tropical countries across three oceans; from South America and West Africa through East Africa and the Persian Gulf, to the Indo-Pacific and Pacific Islands (see Methods). Further, image analysis using Google Earth

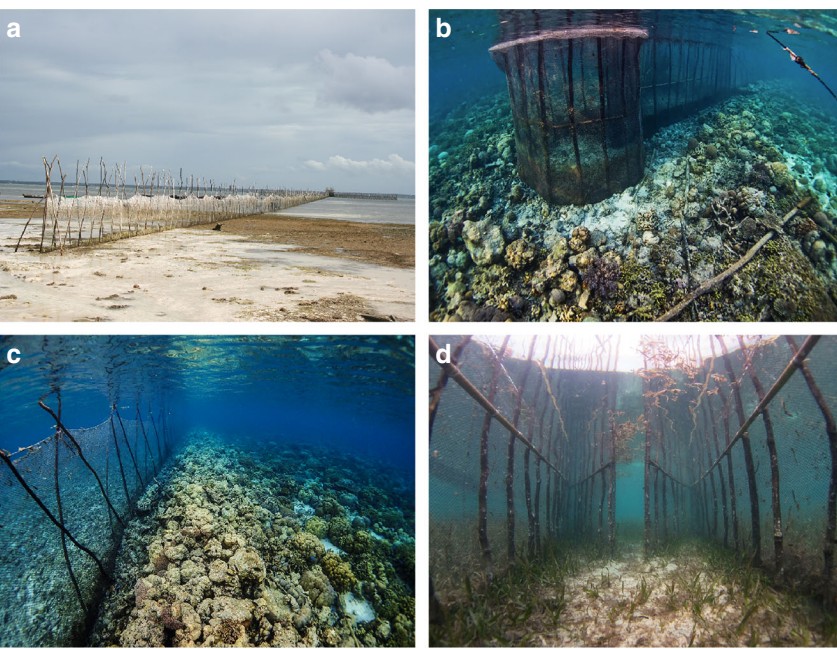

**Fig. 1** Indonesian fish fences on intertidal reef flats, showing **a** an 80 m long fence at low tide, **b** a halo of benthic habitat loss caused by direct clearance and/or the use of poison, **c** a fence acting as an artificial barrier preventing a healthy benthic community from persisting on the landward side and **d** cleared seagrass within the fence structure. Photo credits: Benjamin Jones, Project Seagrass

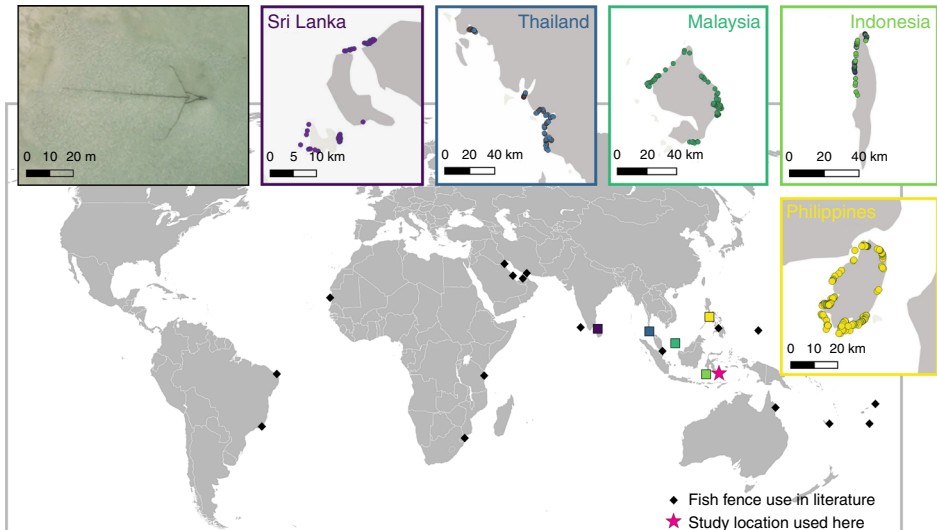

**Fig. 2** Global distribution of artisanal fish fence use, showing the location of mentions in the literature (black diamonds), Google Earth visual surveys showing intensive use of fish fences (inset maps corresponding by colour to squares on main map), study site location used for this study in Wakatobi National Park, Indonesia (pink star), and an example fish fence viewed via Google Earth (inset photo). Where literature mentions describe regional studies with ranges of hundreds of kilometres, symbols denote the approximate centre point of those studies. Map data are from Google

(based on the latest satellite imagery available in July 2018) and the personal observations of the authors, identified their use in three additional countries (Sri Lanka, Thailand and Malaysia), as well as locations in Philippines and Indonesia (Fig. 2). At each location, fence use was shown to not only be present but also in high density.

Long-term data from Kaledupa Island (Wakatobi National Park, Indonesia) show an increase in artisanal fish fence effort over a 15-year period (Fig. 3a; Supplementary Table 1). From just 37 fences in 2002, numbers increased by 450% in only 7 years, peaking at 210 in 2009. Fence length (based on the total length of the central spine) showed a similar trend (analyis of variance; ANOVA, $F_6 = 9.00$, $p < 1.25^{-7}$), increasing by 69% in 11 years from $106 \pm 9$ m (mean $\pm$ 1SE) in 2005 to $179 \pm 9$ m in 2016. Finally, mesh size used in the holding structure almost halved (Kruskal–Wallis, $H_5 = 64.42$, $p < 1.48^{-12}$) from $2.54 \pm 0.00$ cm in 2005 to only $1.28 \pm 0.43$ cm in 2016. If all fences were placed end-to-end, they would have stretched for a total of 10.57 km in 2005, and 27.89 km in 2015. Fences are typically arrow-shaped, consisting of a central spine and multiple wings, and those studied here on average incorporated 2.5 times the length of their central spine in total fence length. This means a combined total of 69.73 km of physical barrier was in place around the island in 2015; significantly exceeding the island's ca. 60 km coastline.

When discussing temporal patterns in total fence effort, it is important to note that, owing to the large spatial footprint and semi-permanent nature of this gear type, both the size and number in use will ultimately be limited by space availability. This could create a false impression to managers that total effort is stabilising, whereas it could simply indicate that spatial saturation is being reached, and that effort may continue to rise through regular mesh size decreases as seen here.

Fishers on the island appear to have changed their main fishing gear preferences, moving from widespread use of low-intensity methods (lines and spears) to a greater reliance on net fishing and fish fences. As part of stratified randomised household interviews throughout Kaledupa, 209 fishers were interviewed in 2005, and 75 interviewed in 2012. The majority of fishers in 2005 used lines as their main gear (54%), with 21% using nets and 7% using fish fences. Ten per cent used small mobile fish traps (bubus) and 5%

used spears. By 2012 the majority of fishers used nets (62%) with 30% now using fish fences as their main gear. In 2012, no one reported using spears or lines as their main gear and one fisher reported using a fish attraction device.

Despite this widespread and growing use and large cumulative spatial footprint, fish fences are still only of direct economic benefit to a small proportion of the total fishing community. Each fence is typically owned by a single fisher, and no additional labour is required to maintain their catches unless the owner chooses to employ an assistant to collect catches on their behalf. The widespread negative impacts of fish fence use must be placed in the context of their status as a minority gear type to fully appreciate the mismatch between the economic benefits to a few and the ecological costs suffered by all.

**Impacts within and between ecosystems**. Local expert workshops and household interviews conducted between 2011 and 2014 revealed the presence of a range of direct physical impacts caused by the construction and use of artisanal fish fences. Considering the importance of habitat in supporting fish stocks[31], the loss of habitat from these direct impacts will expand the footprint of fish fence use beyond the simple removal of fish by reducing local carrying capacity and recruitment potential. Fishers commonly remove all seagrass within the immediate vicinity of the fence (Fig. 1d), either through cutting or the use of poison as it is believed that this enhances catch. This creates a patch of habitat loss within the structure of the fence, as well as a halo surrounding it (Fig. 1b), resulting in habitat fragmentation and edge effects, which are known to have negative ecological impacts[32,33]. Fences also introduce an artificial physical barrier, acting as an obstacle to natural ecosystem functioning, which will interrupt the natural movement of mobile species, but will also impact hydrology, connectivity, and even infochemical processes[34], which, in turn, can disrupt benthic community structure. This can lead to a healthy habitat coming abruptly to an end at the seaward side of a fence, to be replaced by a barren seascape on the landward side (Fig. 1c).

Fences are typically constructed using wooden poles harvested in local mangroves, further increasing the ecosystem scale impact of these fisheries. In the Wakatobi, *Bruguiera gymnorhiza* wood is

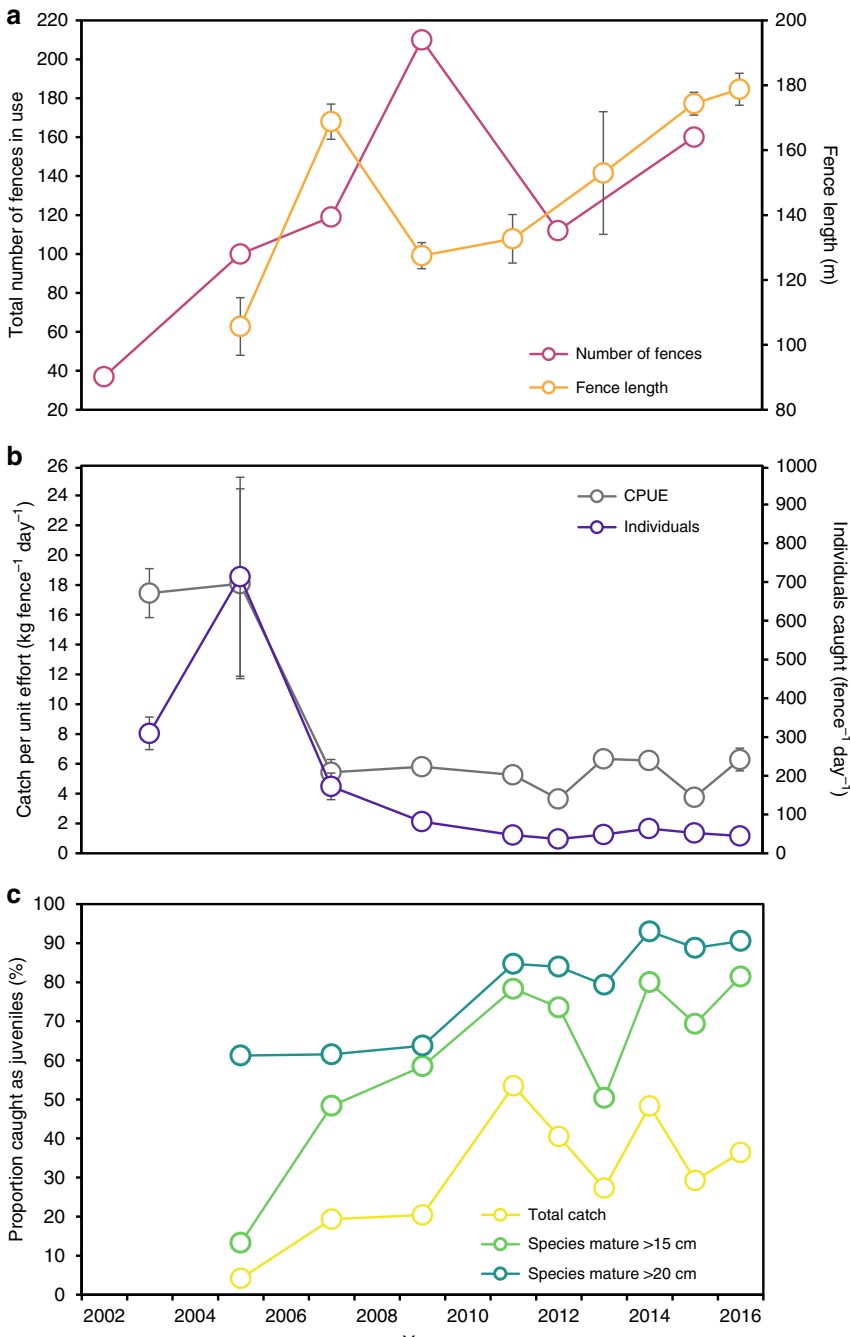

**Fig. 3 a** Total fish fence effort around Kaledupa Island between 2002 and 2016, showing total number of fish fences in use (pink), and length of fences measured along the central spine (orange). **b** Catch statistics from fish fences between 2003 and 2016 during intensive 5-week study periods July–August, showing catch per unit effort (CPUE; grey) and number of individual fish caught (purple). **c** Overall proportion of fish caught as juveniles for the total catch (all species combined; yellow), for species that mature above 15 cm (green) and for species that mature above 20 cm (blue). Data shown in **a** and **b** are mean ± 1 standard error (SE) where applicable. For *n* of fences and catches surveyed at each time point see Supplementary Table 1

used[35], usually with a 2 m long pole every 0.5 m along the entire length of fence. Based on each structure incorporating on average 2.5 times the length of its central spine in total fence length, this results in ~ 500 poles being used for the construction of a 100 m long fish fence. Using the total number and mean length of fences in 2015, this equates to 139,464 poles being used. Assuming a mean pole radius of 2 cm, and using the average of published wood density values for *B. gymnorhiza*[36], we estimate 268 tonnes of mangrove wood was required to construct the fences present around Kaledupa in 2015, representing a substantial driver of local mangrove deforestation.

Long-term catch monitoring suggests severe and far-reaching ecological impacts from intensive artisanal fish fence use. At the broadest level, catch per unit effort (CPUE) declined by almost 70% between 2005 and 2007 (ANOVA, $F_8 = 13.53$, $p < 2^{-16}$), from $18.08 \pm 6.36$ kg fence$^{-1}$ day$^{-1}$ to $5.44 \pm 0.86$ kg fence$^{-1}$ day$^{-1}$ (Fig. 3b). This has subsequently stabilised, fluctuating between lows of $3.65 \pm 0.22$ kg fence$^{-1}$ day$^{-1}$ (2012) and $6.29 \pm 0.76$ kg fence$^{-1}$ day$^{-1}$ (2016). The number of individual fish caught showed a similar decline of over 75% (Negative Binomial generalised linear models (GLM), $F_8 = 478.18$, $p < 2.2^{-16}$), from $713.50 \pm 256.67$ individuals fence$^{-1}$ day$^{-1}$ in 2005 to $172.76 \pm$

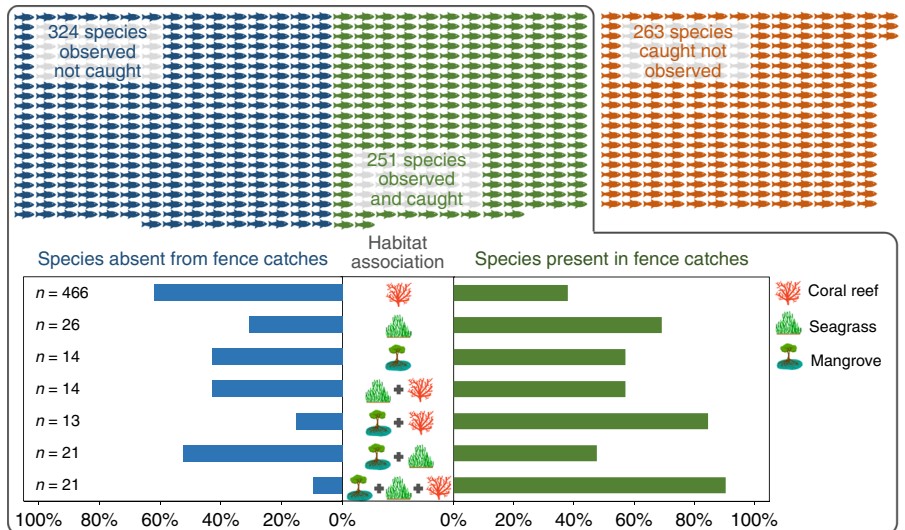

**Fig. 4** Visual representation of the 838 fish species recorded locally, separated by their presence in visual surveys only (blue fish), fish fence catch monitoring only (red fish) or both (green fish). All 575 species recorded in visual surveys (inside 'box') were then allocated habitat associations, based on which visual surveys they were recorded in (coral reef, seagrass, mangrove). For each habitat association, the proportion (%) of species present in (blue bars) and absent from (green bars) fish fence catches are shown, with the total number of species allocated to that habitat association (*n*). Symbol artwork produced by Olivia Farman, Operation Wallacea

34.32 individuals fence$^{-1}$ day$^{-1}$ in 2007. However, unlike CPUE, this downward trend continued, and by 2012 only $36.80 \pm 3.10$ individuals fence$^{-1}$ day$^{-1}$ were being caught. Considering the increases in average fence size and decreases in mesh size during this time, the true extent of decreasing returns from fish fences will be significantly higher than reported here. Stabilised CPUE should not be confused with stable returns. There was also a > 30% decline in the species diversity of catches observed here (ANOVA, $F_8 = 17.29$, $p < 2^{-16}$), which decreased from $20.00 \pm 1.72$ species catch$^{-1}$ in 2005 to $13.69 \pm 0.59$ in 2016, presumably as species become locally rare or even extirpated.

Over the course of this study, the proportion of juvenile fish in individual catch averages increased by > 400%, from $9.72 \pm 3.33\%$ in 2005 to $40.27 \pm 1.92\%$ in 2016 (ANOVA, $F_8 = 19.81$, $p < 2^{-16}$). For species that mature > 15 cm, $81.37 \pm 1.73\%$ were caught as juveniles in 2016 compared to $40.79 \pm 9.02\%$ in 2005 (Quasibinomial GLM, $F_8 = 148.47$, $p < 2.2^{-16}$), and those that mature at > 20 cm were caught at $89.73 \pm 1.58\%$ in 2016 compared with $57.63 \pm 9.82\%$ in 2005 (Quasibinomial GLM, $F_8 = 96.12$, $p < 2.2^{-16}$). When all catches are combined, the proportion of juveniles caught each year increased from 4.15% to 36.38% (all species), 13.25% to 81.41% (species that mature > 15 cm) and 61.23% to 90.54% (species that mature > 15 cm) (Fig. 3c). Minimising the removal of juvenile fish is a core principle of the ecosystem approach to fisheries management[37], whereas larger species tend to be the most ecologically and commercially valuable. Their loss can have significant detrimental impacts, including cascade effects from predator declines[38] and macroalgal overgrowth from large herbivore loss[39], and so the removal of almost exclusively juveniles will likely have a devastating impact on their survival prospects, and potentially on ecosystem resilience as a whole[40].

Analysis of the 10 most abundant species found in catches across years showed an overall decrease in median lengths from the start to the end of this study for seven species; two of these decreases were significant (Moods Median Test; Supplementary Table 2). Interestingly, 2 of the 10 species showed significant increases in median length over the course of the study: *Lethrinus rubrioperculatus* and *Siganus canaliculatus*. On one hand this could indicate that, in such a complex multi-species fishery, no

individual species were caught in sufficient numbers to accurately detect the true length distributions at a single time point. Alternatively, if these changes are indeed accurate, it could suggest that some species are benefiting from the intensive removal of competitors/predators elsewhere in the system (e.g., from other gear types), or they represent potential winners of environmental change. For example, *S. canaliculatus* feeds on epiphytic algae, which tends to proliferate with habitat degradation.

Fish surveys via underwater visual census (UVC) on adjacent coral reefs show a corresponding decline in overall abundance. When all sites and depths are combined, fish densities (individuals 250 m$^{-2}$) almost halved in 10 years, from $910 \pm 88$ in 2002 to $498 \pm 31$ in 2012 (Supplementary Figure 1). This was largely driven by a loss in upper outliers, representing the most-pristine reef areas, with maximum densities encountered falling over the same time from 2437 to 1196 individuals 250 m$^{-2}$, and range declining from 2214 to 969. Whereas reef fish density decreased negatively with the number of fish fences in use (Supplementary Figure 1), this was found not to be statistically significant (linear regression $p = 0.08$), although in a complex multi-gear fishery this is unsurprising. However, there is evidence to suggest that fish fences are an important, and under-appreciated, driver of these ecosystem wide declines, namely (i) their high exploitation of vulnerable life stages, (ii) their non-selective nature and subsequent exploitation across a diverse fish community, and (iii) their inherent purpose of disrupting the natural movements of fish populations that limits vital connectivity.

Owing to the semi-permanent nature of artisanal fish fences, combined with their large dimensions and small mesh size ($1.28 \pm 0.43$ cm in 2016), they are highly non-selective (i.e., they remove a large proportion of the available community including early life stages). When this is framed within the hyperdiverse marine ecosystems of the Indo-Pacific[12], the footprint of such non-selective gears across the entire fish community can be huge. Fish community composition around Kaledupa is well known, thanks to over two decades of intensive research combining visual surveys and fishery catch monitoring (see Methods). A total of 575 fish species have been recorded by visual surveys of local reef,

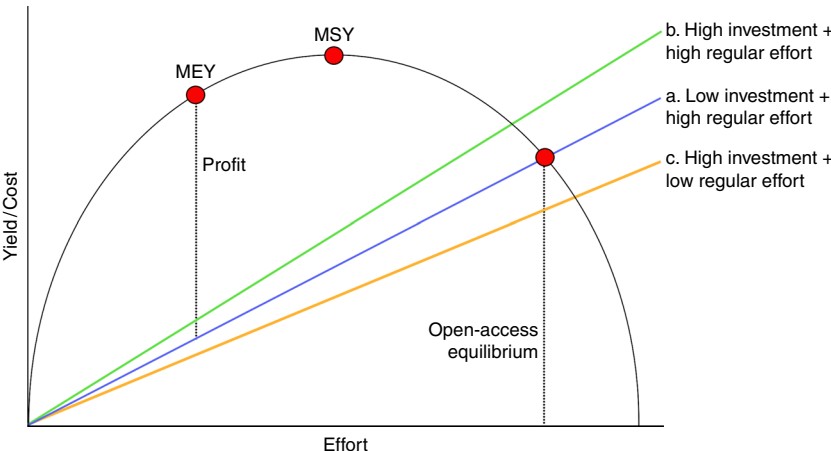

**Fig. 5** The Gordon-Schaefer fixed price bioeconomic model[75] indicating the open-access equilibrium where cost equals the effort–yield curve. Also shown are theoretical variations of this relationship based on differing levels of initial investment and regular effort. The high-investment–low-effort nature of fish fences could exacerbate overfishing by removing some traditional bioeconomic barriers

seagrass, and mangrove habitats historically. Of these, 251 (44%) have also been recorded in fish fence catches (Fig. 4). However, a further 263 species have been recorded in fish fence catches that have not also featured in local surveys, meaning these fences are known to exploit at least 514 fish species to some extent at a single location. This includes charismatic species such as sharks and rays, which were observed as by-catch during this study in addition to non-fish species including turtles and various invertebrates. Although discarding of by-catch was minimal during this study, it has recently been recorded in other parts of the Indo-Pacific[41], suggesting further sustainability concerns for artisanal fisheries.

Fish fences are designed to take advantage of the natural movements of fish, by forming a barrier from which they are unable to escape as the water recedes towards low tide. This causes disruption to the well-described, and ecologically important, natural connectivity between coastal marine habitats[42,43]. When the cumulative extent of this barrier is equivalent to ca. 50% of the coastline, as seen here around Kaledupa Island, this disruption is likely to be extreme. When the species recorded in local surveys are categorised into habitat associations based on the specific visual surveys in which they have been recorded locally, the proportion of which are known to be exploited by fish fences increased in those communities occupying multiple habitats (Fig. 4). For example, 90% of the species observed in "reef + seagrass + mangrove" and 85% of species observed in "reef + mangrove" have been recorded in fence catches. This is compared with only 38% of species observed only on reefs. Natural connectivity enhances fish abundance and protects ecological processes[19], as well as preserving resilience[18], and so any activity that acts to disrupt this connectivity will be counterproductive to conservation efforts. Considering reef fish have been shown to migrate over 30 km[44] even local fish fence use can have far-reaching implications across a wide geographical area.

**Bioeconomic controls and social conflict.** Indonesian fishers will typically invest ca. USD 400 (correct as of 2018) of capital to construct a new fence, and typically no further financial outlay is required for at least 1 year, after which there will be small costs associated with maintenance and repair. This level of capital investment is similar to other gear types used on coral reefs, e.g., gill nets[45], but while these alternatives are also typically associated with high regular effort (labour at sea and on land), fish fence use

requires little regular effort from the fisher for as long as the structural integrity of the fence is maintained (at which point they can choose whether to repair, replace or exit the fishery). Fish fences are generally moved up to four times per year. Daily effort is restricted to a single trip to empty the fence at low tide and return the catch to land. This makes artisanal gears an example of a highly efficient gear, which are known to impact fish stocks[46], but also means fence owners are forced (by the need to recover capital investments) and incentivised (by long-term cost neutrality and low regular effort) to operate their fences for as long as possible. Once a fence is constructed, there exists a lack of any economic or energetic barriers to overfishing. This is coupled to a lack of any technical barrier in the form of gear availability, access to fishing grounds, and required expertise. Thus, fishing pressure is maintained in the face of diminishing returns and habitat damage, satisfying the criteria of Malthusian overfishing when framed alongside continued high human population growth in the tropics[47,48].

From a bioeconomic perspective, increasing the cost of fishing (for example, via high capital investment) should theoretically improve sustainability by reducing the total effort at which open-access equilibrium is reached[49]. However, conversely, a reduction in regular unit cost/effort should have an opposite effect, exacerbating overfishing beyond the original open-access equilibrium. The passive nature of fish fences could therefore negate the potential benefit of capital investment as a control mechanism against overfishing, and could even accelerate fishery collapse (Fig. 5). This impact is worsened by the privatisation of fish fences in recent decades, having evolved from cooperative to single ownership. This concentrates the benefits of fence use within a narrow group of people, and displaces others into other gear types to maintain livelihoods, thus intensifying overall fishing pressure. It is also likely an important contributing factor toward the increase in fence numbers documented during this study as former cooperative members switch to individual ownership.

Many tropical marine habitats, including those studied here around Kaledupa Island, are open-access[50]. Fisheries governance can be strengthened by the addition of property rights such as territorial user rights fisheries[22], or via catch shares in the form of quotas[51], in order to avoid a tragedy of the commons scenario[52]. However, fish fence owners assume perceived property rights while excluding other stakeholders from accessing traditionally open-access fishing grounds. Fish fence owners then assume an

unofficial (and unregulated) exclusive spatial harvest right, while undermining the catches of the wider stakeholder community through previously discussed disruption of ecological connectivity and high juvenile removal. The fences surveyed as part of this study averaged a 5:1 length to width ratio, meaning that in 2015 (the most recent year in which all effort metrics are available), a mere 160 fishers (one fisher per fence) assumed dedicated access privileges over an estimated $0.97 km^2$ of fishing grounds in addition to the ca. 70 km of physical barrier limiting the natural movement of fish stocks.

Fish fences also create social conflict in Indonesia owing to perceived social hierarchy between *Pulo* (islander) and *Bajo* (sea nomad) communities. Informal restrictions mean only *Pulo* fishers are socially permitted to own fences, therefore excluding not only based on economic status but also on ethnicity. To our knowledge, this is locally unique to fish fence ownership, with no other gear types restricting *Bajo* involvement based on social hierarchy alone, although they are often marginalised by a lack of financial capacity to utilise more technology-driven fishing techniques. When combined with the issues surrounding assumed property rights and privatisation, fish fences undermine co-management efforts, which rely on buy-in across the stakeholder group[53]. Our interviews with local fishers also reveal evidence of increasing levels of conflict between fishers over space, as catches decline and fishers look to move into new areas.

**A new management priority?**. Considering the widespread shortfalls in resources impacting the performance of marine conservation interventions[17], it is crucial that management effort is directed to maximise positive change via evidence-based recommendations[54]. In the case of complex tropical multi-gear fisheries, a better understanding of the impact and extent of individual gear types will help identify those having a disproportionate impact and make more informed decisions. For example, it can help ensure that no-take areas (NTA) are matched by evidence-based management of gear types used outside of them[15,16,20,55]. Fish fences are used where fish protein forms a major part of people's diets;[10] thus maintaining sustainable fisheries is a high priority to protect human health, livelihoods and culture. Yet, they disrupt vital ecological connectivity, remove high quantities of juveniles, exploit hundreds of species, lack traditional economic and social barriers to overfishing and create social conflict among wider stakeholder groups, whilst only benefiting a minority of the overall stakeholder community. This suggests a worrying, and disproportionately high, impact on both fisheries sustainability and the multiple ecosystems on which they depend.

Restricting fish fence use would represent a low-effort—high-reward conservation strategy as they are semi-permanent and thus easily detectable, making them far easier to police than many other gears. This effort would result in direct and immediate benefits to the health and extent of three distinct ecosystems (coral reefs, seagrass beds, and mangrove forests), would remove a significant sink for large numbers of juvenile fish across many hundreds of species, and would benefit the wider fishing community by removing an important source of social conflict whilst re-opening large areas of currently inaccessible fishing ground and restoring natural fish movement/migrations. There would likely be displacement into alternative gear types, but owing to the wide-ranging threats from fish fence use outlined here, we would anticipate this to have an overall net benefit to the fishery as a whole. Without restriction, the low regular effort required to maintain operation combined with the economic need to recover initial investment will likely mean their widespread and intensive use persists regardless of further

declines in fish stocks. We therefore call on managers to target gear restrictions toward fish fences as an urgent priority in achieving sustainability in these vital fisheries, while improving our understanding of the hidden threats of artisanal fisheries by applying our approach to a broader suite of gear types.

## Methods

**Satellite imagery visual analysis to determine extent of artisanal fish fence use**. Owing to the large and semi-permanent nature of artisanal fish fences, their structures are easily visible from satellite imagery[30]. Artisanal fish fences have previously been mentioned within the scientific literature as being used in 19 countries within the tropics: Indonesia;[23,25,56] French Polynesia;[57] Micronesia;[58] Samoa;[59,60] Tonga;[61] Mozambique;[62] Tanzania;[25,63] Kuwait/Saudi Arabia/Qatar/ United Arab Emirates/Iran/Bahrain;[30] Brazil/Mauritania/India/Singapore/Philippines/Australia[25]. Here, we use Google Earth to identify their widespread use at locations in three additional countries (Malaysia, Thailand and Sri Lanka), plus new locations in Indonesia and Philippines, to better highlight the geographical scale of their use. These locations were selected based on personal field observations of the authors, and for each location visual scans were performed, with the GPS co-ordinates of each visible fence recorded (Fig. 2). Visual scans were repeated twice by the same researcher to ensure accurate counts.

**Longitudinal case study location**. The Wakatobi Marine National Park, Southeast Sulawesi, was designated in 1996 and covers 1.39 million hectares. Kaledupa is the second largest island in the park, with a human population of around 17,000 distributed among 27 villages with a high reliance on artisanal fisheries[23]. There are two distinct ethnic groups present on and around Kaledupa: the land-based *Butonese* descendents (*Pulo*) and a nomadic sea people (*Bajo*) who settled in permanent villages built on artificial platforms over the reef flats during the 20th Century[64]. Owing to assumed ownership rights over fishing grounds locally, they are only constructed and owned by *Pulo* fishers.

**Local effort assessment**. To estimate the effort allocated to artisanal fence use by the local fishing community during the course of this study, three values were quantified. First, island-wide visual censuses of fences were conducted by circumnavigating Kaledupa and its outlying islands by boat. A similar pre-study census from 2002 was also included as a historical time point (unpublished). Second, the total length of each fence monitored in this study was measured along the central spine (*Panaju*) to provide an approximate mean total fence length for Kaledupa. The mesh size in use in each fence was also measured in the collection end (typically, the finest mesh size in use within the structure). Census and fence length data are unavailable for all years owing to logistical constraints, specifically adverse weather conditions prohibiting island-wide access.

**Household surveys and workshops**. To understand marine and coastal resource use patterns around Kaledupa Island, a series of stratified randomised and semi-quantitative household interviews were conducted around the Island (spread across 17 villages) in both 2005 and 2012 (see ref. [65]). Within these interviews 209 fishers were interviewed in 2005 and 75 interviewed in 2012, and the data specific to fishers were examined in both years with respect to fisheries gear use.

In order to gain detailed insights into the methods and broader environmental context of fish fence use, we ran two local expert focus discussion workshops (see ref. [66]) during 2012. One of these was with the local government, NGO, and village-level officials who had involvement in fishery activity. The second workshop was with fishers from the local indigenous *Bajo* communities. Discussions focussed on perceived impacts on the local marine environment and long-term ecological change. Ethical approval for working with human participants was obtained from Swansea University (SU-Ethics-Staff-250319/134), and informed consent was obtained.

**Catch monitoring**. Catches from artisanal fish fences were monitored around the northeast coast of Kaledupa, in 5-week-long intensive sampling periods between July and August in 2003, 2005, 2007, 2009, and annually between 2011 and 2016. During each sampling period a locally run fish fence cooperative (known locally as a *Kelompok*), comprising between 7 and 20 individual fences and their owners, was selected at random and their owners approached informally to discuss participation in the study. Each fence was emptied daily, as usual at low tide, and the catch returned by the owner to a centralised location (typically an owner's home) in the village of Laulua (5°29'56.08"S 123°44'52.97"E) for analysis.

On arrival, the total catch was weighed to provide CPUE, reported as kg fence$^{-1}$ day$^{-1}$. All fish were subsequently identified to species level, and length measurements taken of all individuals. Where > 20 individuals of a species appeared in a single catch, a random sub-sample of 20 individuals were chosen for length measurements and mean values applied to the total number caught. Length data were used to quantify the proportion of each species and total catch caught as juveniles using published species-specific size of maturation values[67]. Where these values were not available, a value one-third of the species-specific maximum size

was used to estimate minimum size of maturation[68,69]. To explore trends in the length of caught species, 10 species were chosen that were caught in the highest numbers throughout the study (excluding the smallest shoaling species, e.g. sardines). The median length of each of these species was then calculated from the catch monitoring data.

Over the nine sampling seasons, 995 catches were surveyed comprising 70,967 individual fish. As no animals were harmed for the purposes of this study (normal daily catches of a working fishery were observed), ethical approval was not sought.

**Fish surveys of adjacent reefs.** Densities of fish on adjacent coral reefs were determined via a reef monitoring programme between 2002 and 2012. During this time, six reef sites were visited between June-August in 2002–03, 2005, 2007–09, and 2011–2012. UVC[70] was performed along $50 \times 5 \times 5$ m belt transects, with all fish identified to species level. At each site and year, triplicate transects were completed on three reef zones: reef flat (0–3 m), reef crest (3–8 m), and reef slope (8–15 m).

**Habitat associations.** Species lists from fish fence catch and reef fish surveys were combined with similar lists from published surveys of Kaledupan seagrass and mangrove habitats[43,71–73]. Each species was then assigned a particular habitat association (e.g., reef + seagrass + mangrove), based on the surveys in which it was recorded. These data provided a casual estimate of the inter-habitat connectivity present for each species locally, and the proportion of each community known to be exploited by fish fences.

**Statistical analysis.** CPUE and reef fish survey data were log-transformed and data on species diversity and the total proportion of juveniles square-root-transformed to allow parametric statistical testing. Significant variations in the means of the above variables over time, along with untransformed data on fence length, were tested for using one-way ANOVA, with the specific source of significance explored using post hoc Tukey–Kramer testing. Variations in mesh size were tested for using non-parametric Kruskal–Wallis with pairwise Mann–Whitney $U$ with Bonferroni correction. Data on the number of individuals per catch, as well as the proportion of juveniles amongst species that mature over either 15 cm or 20 cm body size failed to conform to the assumptions of linear regression, and so negative binomial (individuals) and quasibinomial (proportion of juveniles among larger species) GLM were performed alongside post hoc Tukey–Kramer testing. Differences in fish length for individual species between the start and end of the study were performed using Moods Median Test. All statistical analyses were performed using R[74].

**Reporting summary.** Further information on research design is available in the Nature Research Reporting Summary linked to this article.

## Data availability
The data sets presented in this study are available from the corresponding author on reasonable request. The source data underlying Figs. 3 and 4, Supplementary Fig. 1 and Supplementary Tables 1 and 2 are provided as a Source Data file.

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

## Acknowledgements

This research was funded by Operation Wallacea in partnership with Hasanuddin University, Makassar, with permission from RISTEK. We thank all field scientists and student volunteers who helped with data collection, and thank the local communities of Kaledupa Island, in particular the fish fence owners themselves, for allowing their daily routines to be disrupted in order for our surveys to be conducted. We are particularly grateful to Beloro and his team at the local Kaledupan fisheries NGO FORKANI, who provided invaluable local knowledge and facilitation; this project would not have been possible without them. We also thank Pippa Mansell for logistical assistance and fish fence census data collection.

## Author contributions

DAE and DJS conceived the study and JJ advised on study design and local requirements. Catch-monitoring data were collected by DAE, GNA, DM, JR and PWS, household survey data were collected by LCC-U and RKFU, and visual survey data collected by RKFU and DJS. DAE wrote the manuscript with extensive input from all other authors.

## Additional information

**Competing interests:** The authors declare no competing interests.

