## [Peer Review File · Nature Communications]

Reviewers' Comments:

Reviewer #1:

Remarks to the Author:

Understanding the multiple stressors that influence multi-species reef fisheries is an incredibly important goal, and the authors here present a clear example of how a specific type of fishery in a specific location can illustrate the threat of fish fences. The work from Indonesia demonstrates that these fences can cause shifts in fisheries including a worrying increase in juveniles. Additionally they also highlight the indirect impacts on the ecosystem that this fishery entails, including deforestation of mangroves and shifts in power relations among the fishing community.

Where I think the paper falls short is extrapolating from this individual site to a global phenomena. I may be reading the MS wrong, but I feel like the introduction is written to talk about fish fences as a global phenomena whereas the data is from one site in Indonesia. These are useful data, but I'm not sure if they are generalizable enough for the readership of Nature Communications. I think this would be better suited for a mid-range conservation journal like Oryx or, if the authors want to stay in the Nature family, Scientific Reports is a great home for papers like this which have solid science.

I do have line by line comments below:

Line 64, I see the point you're making but that number seems high (I'll check reference)

Line 75 Move the definition from line 90 to here.

Line 103 These data seem out of place as you try to build your case. I think it would be better to integrate them into a global summary or synthesis

Line 130 Did you also see a subsequent change in fish targeted or size classes of fish?

Line 149 is it really an open environment? My understanding is that many fish fences are set up around existing channels and passes? Perhaps I'm wrong, and I see where you're going, but I feel like this is a bit overstated.

Line 153 Please cite an example of fish fencing impacting benthic community structure.

Line 165 Significant has a specific meaning, which I don't think you mean here (e.g. no statistical test has been applied), please reconsider and/or place in context. I don't doubt that this is a negative impact but this statement needs to be supported.

Line 180 Did other fisheries see a similar decline? Place these data in a larger context, was it just the fences doing poorly or was the whole fishery being degraded. One piece of data that would be helpful would be to look at the average length of the species that are remaining; one would assume that the average median length of the species in 2016 would be smaller than those in 2005

Line 186 perhaps report the median fish densities over time?

Line 194, I'd move this paragraph up to after the line 180 paragraph as it supports how the catch has changed over time

Line 209 how small exactly is the mesh size?

Line 264 do we see an increase in fences b/c formally communally operated fishing groups are no each working individually as fence owners?

Line 292 but did other forms of fisheries encourage participation across ethnic lines as well?

Line 321 The main thrust of this paper is in Indonesia, so I don't really see why these other data are included. I would just remove them and streamline the story

Line 368 if the Bajo communities are not included in fence ownership, why were their interview data included here?

Reviewer #2:

Remarks to the Author:

Exton and colleagues propose a study of the impact of fish fence in tropical marine ecosystems. I initially thought that this was another impact study of a fishing gear. However, it is not. The study is incredibly powerful. The authors use an amazing dataset and explore many if not most facets of the impacts of fish fence - ecological, social, and economic. They show that this gear type has an incredibly strong impact on three major interconnected ecosystems (coral reefs, mangroves, seagrass) but also creates social issues and is an economic non-sense. They also show the spatial extent of the use of this gear type across three oceans. The paradox is that in most countries, fish fence are considered as traditional methods, and hence assumed to be less impacting than modern fishing techniques. The study of Exton and colleagues clearly shows the opposite: fish fence are one of the most destructing fishing gear in tropical marine environment. I therefore believe that the publication of this work is urgent to let know managers and scientists from many countries.

The data are amazing, the paper very well written, and I could not see any issue in the statistical analysis. It has been a pleasure to read this very interesting study and I think the paper will attract a wide readership.

I only have two minor comments:

- Issues with figure numbering (probably due to moving a paragraph up): page 4, line 16: the first cited figure is fig 3. I think fig 3 should be renamed as fig 1 and cited as fig 1. Page 5, line 1: fig 1 is cited; fig 1 should be renamed fig 2 and cited as fig 2. Page 5, line 4: same for fig 2 to be renamed fig 3; and then pls check fig numbering along the ms.
- Issue with supplementary Fig 1 (I think means are not orange diamonds but blue circles, and transect values small grey circles)

Laurent Vigliola

REVIEWERS' COMMENTS:

Reviewer #1 (Remarks to the Author):

Understanding the multiple stressors that influence multi-species reef fisheries is an incredibly important goal, and the authors here present a clear example of how a specific type of fishery in a specific location can illustrate the threat of fish fences. The work from Indonesia demonstrates that these fences can cause shifts in fisheries including a worrying increase in juveniles. Additionally they also highlight the indirect impacts on the ecosystem that this fishery entails, including deforestation of mangroves and shifts in power relations among the fishing community.

Where I think the paper falls short is extrapolating from this individual site to a global phenomena. I may be reading the MS wrong, but I feel like the introduction is written to talk about fish fences as a global phenomena whereas the data is from one site in Indonesia. These are useful data, but I'm not sure if they are generalizable enough for the readership of Nature Communications. I think this would be better suited for a mid-range conservation journal like Oryx or, if the authors want to stay in the Nature family, Scientific Reports is a great home for papers like this which have solid science.

We would like to thank the reviewer for taking the time to read our manuscript and provide helpful comments and suggestions. We are glad that they view the data presented as useful and the science as solid. However, we feel that our efforts to frame the threats posed by fish fences as a global phenomenon are warranted and worthy of the broad readership typical of Nature Communications. The target audience of this study spreads far beyond the tropical marine conservation community, let alone beyond Indonesia specifically, as we provide a highly novel example of how to utilise interdisciplinary research to examine the impacts of a resource extraction activity. Using a local case study allowed us to explore fish fence use in high resolution and over a long time period. That would have been impossible (at least within the logistical and financial framework available to us) to achieve were we to have included multiple locations. However, the findings from this case study are directly applicable to any location where fish fences are used along tropical coastlines. It was therefore a logical and necessary step to build on our localised case study by exploring the extent of fish fence use across three oceans. This allows us to highlight the threat posed by fish fence use, not only at our case study location, but more broadly throughout the tropics.

I do have line by line comments below:

Line 64, I see the point you're making but that number seems high (I'll check reference)

We believe this is partly due to how a distinct gear type is defined, and we have modified the text to clarify. Although gear types do show general broad patterns globally, the diversity of ethnicity and culture in Indonesia means that fishing gears have many variations upon these general types. In general, gears can be grouped into a smaller number of classes (e.g. nets, traps), but within those classes can exist multiple distinct varieties of individual gears that each exhibit unique characteristics. In fact, when approached in this way, studies have reported as high as 67 (Selgrath et al 2018 ICES Journal of Marine Science) and 93 (Selgrath et al 2018 PLoS One) gears being used locally. To help clarify this for readers we have modified the text to read "it is not uncommon for over 50 distinct gear types to be in use at a single location (including multiple varieties of the same gear class)" (Line 65).

Line 75 Move the definition from line 90 to here.

This has now been moved as suggested.

Line 103 These data seem out of place as you try to build your case. I think it would be better to integrate them into a global summary or synthesis

Although we appreciate the switch at this point of the manuscript from a global view of fish fences to a local case study, we feel this is an important transition to build a thorough picture of fish fence effort. The local case study allows much higher resolution data to be presented which illustrates the fact that fish fence have grown in use over recent decades, whilst the effort allocated to individual fences (i.e. fence dimensions, mesh size) has also changed in ways that increase the overall impact of this gear. We feel this builds well on the previous paragraph where we place fish fence use within a global context, using a combination of spatial extent and localised intensity at a selection of locations.

Line 130 Did you also see a subsequent change in fish targeted or size classes of fish?

Please see our response to a later comment where we describe the addition of fish length data to the manuscript.

Line 149 is it really an open environment? My understanding is that many fish fences are set up around existing channels and passes? Perhaps I'm wrong, and I see where you're going, but I feel like this is a bit overstated.

We have found fish fences being used in a range of environments, including existing channels but also on intertidal flats, seagrass beds and reef flats. Having said that, the reviewer raises a valid point that, at least in some instances, the environment cannot always be described as being open. We have therefore removed this statement.

Line 153 Please cite an example of fish fencing impacting benthic community structure.

Unfortunately fish fences are poorly studied, and so no suitable examples exist in the literature to support this statement. Instead, we rely on the personal observations of the authors and photographic evidence in Figure 1 that show clear impacts on the benthic community in and around fish fences, which we clarify in the following sentence where we refer to Figure 1 as evidence.

Line 165 Significant has a specific meaning, which I don't think you mean here (e.g. no statistical test has been applied), please reconsider and/or place in context. I don't doubt that this is a negative impact but this statement needs to be supported.

We thank the reviewer for pointing this oversight. We have now changed "significant" to "substantial".

Line 180 Did other fisheries see a similar decline? Place these data in a larger context, was it just the fences doing poorly or was the whole fishery being degraded. One piece of data that would be helpful would be to look at the average length of the species that are remaining; one would assume

that the average median length of the species in 2016 would be smaller than those in 2005

Temporal fishery catch data at the study location is limited to fish fences, and no data exist for the other fisheries present, although it is likely that there is general fishery degradation across gear types to some extent at least. However, to explore changes in fish length structure, we have added data to the manuscript on 10 of the most commonly caught species. The multi-species nature of fish fence catches, whereby we recorded over 500 species of fish being removed, means a thorough exploration of length data across all species would be a study in itself, and yet we agree with the reviewer that an indication as to whether changes in length have occurred would be an interesting addition. We therefore selected the ten most commonly caught species that were present in high numbers both in the earliest years (2003-2003) and latest years (2015-2016), ignoring the very small shoaling species that are regularly caught. We then compared the median length of these ten species between the two time points, reporting the findings within a new supplementary table, within the text of the results and discussion, and explained within the methods. The majority of species showed decreased median lengths, although this was only statistically significant for two species. Interestingly, two species showed significant increases in median length, and we suggest possible explanations for this within the manuscript.

Line 186 perhaps report the median fish densities over time?

Fish density data are already summarised within the text. A source data file will also be provided with the final submission that provides values for individual time points to support Supplementary Figure 1.

Line 194, I'd move this paragraph up to after the line 180 paragraph as it supports how the catch has changed over time

We thank the reviewer for this suggestion, we have made this change.

Line 209 how small exactly is the mesh size?

We have now added the mesh size in 2015 within parentheses at the end of this sentence.

Line 264 do we see an increase in fences b/c formally communally operated fishing groups are no each working individually as fence owners?

To clarify this point we have added an extra sentence to the paragraph discussing the privatisation of fish fences, reading "It is also likely an important contributing factor towards the increase in fence numbers witnessed during this study."

Line 292 but did other forms of fisheries encourage participation across ethnic lines as well?

This is an interesting and important point. Although we have never explored this in detail, we have not encountered any other gears that discourage participation across ethnic lines, although financial limitations do prevent certain gears from being used by poorer groups within the overall fishing community. To reflect this, we have added the following statement: "Although undoubtedly a marginalised group, often lacking the financial capital to enable more technology-driven fishing techniques, the authors are not aware of other gears where social hierarchy itself plays such a role."

Line 321 The main thrust of this paper is in Indonesia, so I don't really see why these other data are included. I would just remove them and streamline the story

We feel it is vital that the threats from fish fences outlined within this study are placed within the context of their widespread use across such a large geographical area. We use a localised case study to explore the multi-dimensional impacts of fish fence use, highlighting a number of key threats that are directly applicable beyond the study site (e.g. the bioeconomics underpinning fish fence use, their disruption of ecological connectivity, their non-selective nature and removal of large numbers of juveniles). This localised approach was necessary to generate high resolution data across such a large time period, but the findings more broadly apply to anywhere fish fences are used along tropical coastlines. Considering how poorly studied fish fences are, we felt it was important to include an assessment of how widely used they are, both spatially (i.e. via our assessment of the literature) and based on intensity (i.e. via our visual assessment of satellite imagery). We believe that removing this element of the manuscript would weaken the study unnecessarily.

Line 368 if the Bajo communities are not included in fence ownership, why were their interview data included here?

Although the Bajo communities do not own fish fences, they are an important component of the overall fishing community who possess a remarkable knowledge and understanding of the marine environment. The Bajo fishers have also historically utilised the exact locations of fish fence placement as their fishing grounds. We therefore felt it was important that their views and experiences be included within this study, in particular their views on some of the ecological impacts of fish fence use. We also felt it was important to include Bajo fishers within our household survey component so that the results indicated a true proportion of the entire fishing community (i.e. and not the fishing community minus the Bajo).

Reviewer #2 (Remarks to the Author):

Exton and colleagues propose a study of the impact of fish fence in tropical marine ecosystems. I initially thought that this was another impact study of a fishing gear. However, it is not. The study is incredibly powerful. The authors use an amazing dataset and explore many if not most facets of the impacts of fish fence - ecological, social, and economic. They show that this gear type has an incredibly strong impact on three major interconnected ecosystems (coral reefs, mangroves, seagrass) but also creates social issues and is an economic non-sense. They also show the spatial extent of the use of this gear type across three oceans. The paradox is that in most countries, fish fence are considered as traditional methods, and hence assumed to be less impacting than modern fishing techniques. The study of Exton and colleagues clearly shows the opposite: fish fence are one of the most destructing fishing gear in tropical marine environment. I therefore believe that the publication of this work is urgent to let know managers and scientists from many countries.

The data are amazing, the paper very well written, and I could not see any issue in the statistical analysis. It has been a pleasure to read this very interesting study and I think the paper will attract a wide readership.

We are absolutely delighted that Dr Vigliola enjoyed our manuscript, and are truly humbled by his incredibly positive comments! It is very reassuring to know that the messages we were trying to convey within the manuscript have come across as we intended, and we hope that other readers enjoy it as much.

I only have two minor comments:

- Issues with figure numbering (probably due to moving a paragraph up): page 4, line 16: the first cited figure is fig 3. I think fig 3 should be renamed as fig 1 and cited as fig 1. Page 5, line 1: fig 1 is cited; fig 1 should be renamed fig 2 and cited as fig 2. Page 5, line 4: same for fig 2 to be renamed fig 3; and then pls check fig numbering along the ms.
- Issue with supplementary Fig 1 (I think means are not orange diamonds but blue circles, and transect values small grey circles)

We thank Dr Vigliola for pointing out these mistakes, which are likely leftover from previous versions of the manuscript. We have now ensured figure numbers are in the order they first appear within the manuscript and that they correspond to the correct figures throughout. We have also corrected the error in the legend of Supplementary Figure 1.

Laurent Vigliola